# Cytokines and Onchocerciasis-Associated Epilepsy, a Pilot Study and Review of the Literature

**DOI:** 10.3390/pathogens10030310

**Published:** 2021-03-07

**Authors:** Melissa Krizia Vieri, An Hotterbeekx, Stephen Raimon, Gasim Abd-Elfarag, Deby Mukendi, Jane Y. Carter, Samir Kumar-Singh, Robert Colebunders

**Affiliations:** 1Global Health Institute, University of Antwerp, 2610 Antwerp, Belgium; an.hotterbeekx@uantwerpen.be; 2Molecular Pathology Group, Laboratory of Cell Biology & Histology, Faculty of Medical and Health Sciences, University of Antwerp, 20610 Antwerp, Belgium; samir.kumar-singh@uantwerpen.be; 3Amref Health Africa, Juba P.O. Box 30125, South Sudan; stephenraimon@gmail.com; 4Academic Medical Center, Global Child Health Group, Department of Paediatrics and Department of Global Health, University of Amsterdam, 1105 Amsterdam, The Netherlands; gasim4u83@gmail.com; 5Amsterdam Institute for Global Health and Development, 1105 Amsterdam, The Netherlands; 6Institut National de Recherche Biomédicale, Av. De la Démocratie N°5345, Kinshasa 1197, Congo; debymukendi@yahoo.fr; 7Centre Neuro-Psycho Pathologique, Universite’ de Kinshasa, Kinshasa P.O. Box 127, Congo; 8Amref International Headquarters, Nairobi P.O. Box 27691−00506, Kenya; jane.carter@amref.org

**Keywords:** onchocerciasis-associated epilepsy, cerebrospinal fluid, cytokines

## Abstract

Neuro-inflammation may be associated with onchocerciasis-associated epilepsy (OAE) but thus far very few immunological studies have been performed in children with this form of epilepsy. In a pilot study we measured the cytokine levels in cerebrospinal fluid (CSF) of persons with OAE from Maridi, South Sudan, and from Mosango, Democratic Republic of the Congo (DRC) and compared these results with cytokine levels in CSF of Africans with non-OAE neurological disorders, and Europeans with epilepsy or other neurological conditions. The following cytokines were studied: IL-6, TNF-α, IL1-β, IL-5, IL-4, IL-13, CCL3 (Mip-1α), VEGF-C, VCAM-1. No cytokine was significantly associated with OAE, although a lower IL-13 level was observed in CSF of persons with OAE compared to African controls. Observed cytokine profiles and neuro-inflammation may be the consequence of long-standing epilepsy, concomitant infections and malnutrition. Ideally cytokine levels should be determined in a prospective study in serum and CSF collected at the time of onset of the first seizures.

## 1. Introduction

Recent studies suggest that nodding syndrome is one of the clinical presentations of an epileptic disorder now called onchocerciasis associated epilepsy (OAE) [1]. OAE is characterised by an onset of epilepsy without any obvious cause between the ages of 3−18 years in previously healthy children in an area of high ongoing or past onchocerciasis transmission [1]. OAE including nodding syndrome disappears when onchocerciasis is eliminated [1,2]. 

Epidemiological studies suggest that the *Onchocerca volvulus* parasite is associated with epilepsy but the pathogenesis remains to be elucidated. The skin snip microfilarial (mf) load in children seems to be the main determinant for the risk of developing epilepsy [3,4]. However, mf are not found in the cerebrospinal fluid (CSF) on a regular basis and have not been shown to penetrate the blood–brain barrier (BBB) [5]. Before the implementation of mass ivermectin administration programs, mf were detected in the CSF of certain individuals with a high level of *O. volvulus* infection [6], but mf have not been detected in CSF in recent studies [7,8]. Another potential mechanism is that *O. volvulus* is able to trigger an immunological mechanism that may cause brain inflammation or auto-immune disease [9]. It has been hypothesized that leiomodin-1 antibodies cross-reacting with *O. volvulus* proteins may be neurotoxic and induce nodding syndrome [9] but this hypothesis has not been confirmed. A post-mortem study of five persons who died with nodding syndrome in Uganda suggested that nodding syndrome is a tauopathy, a type of neurodegenerative disease [10]; However, in a more recent post-mortem study on nine persons who died with OAE, including five with nodding syndrome, tau deposits were not found in the brains of two individuals and very sparse in two other individuals [5]. Tau deposits have also been described in persons with refractory epilepsy [11], and therefore the tau deposits in persons with OAE are most likely the consequence of repetitive seizures. This second post-mortem study showed localized signs of neuro-inflammation characterized by gliosis and features of past ventriculitis and/or meningitis in all but one participant [5]. 

Thus far, only three studies have measured cytokine levels in persons with OAE, including nodding syndrome. In a case control study in an onchocerciasis-endemic area in Kasangulu in the Bas-Congo province of DRC, 12 persons with epilepsy and 13 healthy controls were enrolled [12]. Serum cytokines and chemokines were measured using a Luminex multiplex assay. Epilepsy was associated with high serum levels of IL-17, low levels of IL-1RA, and IL-8 [12]. The mean levels of IL-17 and IL-8 in persons with epilepsy were 2.30  ±  0.98 pg/mL and 3.07  ±  0.79 pg/mL, respectively, compared to 1.54  ±  0.54 pg/mL and 3.92  ±  1.07 pg/mL in healthy controls (*p*  <  0.05). 

In a second case control study from South Sudan, 20 children with nodding syndrome and 10 healthy controls were enrolled [13]. Cytokines were also measured using the Luminex multiplex assay. The authors of this study reported a 85−99% reduction of cytokines IL-1β, IL-2, IL-6, IL-8, TNFα and IFNγ in children with nodding syndrome compared to healthy subjects [13]. However, the results of this study need to be interpreted with caution as in this study extremely high cytokines were also reported in healthy controls. For example, IL-8 levels in healthy individuals were reported as 6840 ± 754 pg/mL compared to only 3.92  ±  1.07 pg/mL in the previous study in the DRC. The IL-8 cytokine levels from the DRC were only slightly lower than IL-8 levels reported in a study among European children (9.06 pg/mL) [14]. We speculate that the high cytokine levels observed in the samples from South Sudan may have been the consequence of problem in the way the samples were collected, stored, transported or processed.

A recent case control study conducted in Uganda compared 154 persons with longstanding nodding syndrome and 154 community controls by measuring plasma levels of cytokines, chemokines and complement activation (C5a) markers using ELISA or a custom MagPix Luminex assay. Cytokine CSF levels of the 154 persons with nodding syndrome were also compared with CSF cytokine levels of 15 Ugandan children in remission from a haematological malignancy [15]. No specific cytokines profiles was observe in the CSF of the Ugandan children. However, they were all receiving symptomatic therapy including antiepileptic drugs and bi-annual ivermectin treatment. 

With the aim of identifying a cytokine profile that could provide further information for pathology studies in persons with OAE, we tested CSF samples of ivermectin-naïve persons with OAE from South Sudan using a select panel of 9 inflammatory cytokines and chemokines, based on pro-inflammatory (Th1), anti-inflammatory (Th2) and vascularisation response (IL-6, TNF-α, IL1-β, IL-5, IL-4, IL-13,CCL3 (Mip1-α), VEGF-C, VCAM-1). IL-4, IL-6, IL-13 are type 2 responses playing a critical role in both inflammation and protective immunity [16]. IL-6 is required for protective immune responses against early filarial infection [17]. TNFα and IL1-β have been suggested as epilepsy markers [18] while VEGF-C and VCAM-1 are vascularization markers

In the current paper, we present the data of our pilot study and compare them with the results of the studies conducted in the DRC and Uganda. 

## 2. Results

We were unable to identify any characteristic cytokine profile in the CSF of persons with OAE. Only a decreased expression of IL-13 in the CSF of persons with OAE compared to African controls was noted, *p* = 0.028 (Figure 1A). IL-6, IL-4 or TNF-α could not be detected, either in cases or controls. A higher level of IL1-β was observed in Africans with non-OAE neurological disorders compared to Europeans with epilepsy, *p* = 0.026 (Figure 1B). No significant difference was observe within the 4 groups for IL-5 and CCL3 (Mip1-α) (Figure 1C,D). We did observe a significant different in the expression profile of VVCAM between African controls and Europeans with neurological conditions but no Epilepsy (Figure 2A). However, we did not observe any difference between cases and controls. In addition, no differences was observed among all the groups for the expression of VEGF-C, *p* = 0.022 (Figure 2B).

## 3. Discussion

In this small pilot study of 13 persons with OAE, we did not observe any characteristic CSF cytokine profile. Nevertheless, we noted a decreased level of IL-13 in the CSF of persons with OAE compared to African controls. A higher level of IL1-β was observed in African controls with non-OAE neurological disorders compared to Europeans with epilepsy. We also observed a high level of VCAM in African controls with non OAE compared to Europeans with neurological conditions. However, given the small sample size of the study and without results from cytokine serum levels it is difficult to interpret these results. 

In the case control study in Uganda, C5a in CSF and CRP in plasma of children with nodding syndrome was elevated compared to controls. Moreover, the plasma level of CRP correlated with disease severity. Complement activation is associated with neurological conditions including epilepsy, psychiatric disorders and neurodegenerative disorders [19,20]. In addition, a decreased level of IL-10, IL-13 CXCL-10 (IP-10), CXCL-13 (BCA-1), CCL-2 (MCP1) and a TNF ligand superfamily member APRIL in plasma was observed [15]. The explanation of these decreased cytokine levels in children with nodding syndrome is unclear but similar decreases in Th-2 type responses have been observed in individuals with prolonged occult *O. volvulus* infection [21] and persons with onchocerciasis with a high mf load [22]. CXCL-10, CCL-2 and mainly CXCL-13 are good predictor of neuroinflammation in CSF of children [23]. 

Levels of CXCL9 (MIG), CCL5 (RANTES), IL13, IL6, TNFα, MMP-9 and INFγ in CSF were comparable between cases and controls. In contrast to our results, IL-13 was not detectable in CSF from children with nodding syndrome. Similar to our study, IL-6 and TNFα was also not detectable in CSF. In contrast to our pilot study and the Ugandan children were al treated ivermectin and anti-epileptic drugs. It has been reported that ivermectin may influence the cytokine profile in persons with *O. volvulus* infection [24]. 

The interpretation of cytokine levels in persons with OAE is difficult because an associated *O. volvulus* infection may also influence cytokine levels. Helminth infections induce the production of type 2 cytokines which lead to both expulsion of the parasite and an inflammatory response. Moreover, elevated cytokine levels have also been observed in different forms of epilepsy [25]. For example, cytokines such as IL-1β and IL-8 have been reported to induce epileptic seizures by activating the cytokine cascade [26].

## 4. Materials and Methods

Persons with OAE were selected among participants from previously described studies in South Sudan [27] and the DRC [28]. Characteristics of the study participants are shown in Table 1. Eleven persons with OAE from Maridi in South Sudan were part of a series of 13 persons with OAE with more than 80 *O. volvulus* mf counts in their skin snips and in whom a lumbar puncture had been performed at Maridi State Hospital. CSF had been stored immediately at -20°C before being shipped using the cold chain to the University of Antwerp, Belgium. At the University of Antwerp, the CSF was examined for the presence of *O. volvulus* mf, and *O. volvulus* and *Wolbachia* DNA but in none was evidence of *O. volvulus* or *Wolbachia* infection found [8].

Two persons with OAE and 17 African controls with non-OAE neurological disorders were selected from the project “Better Diagnosis for Infectious Diseases“ (NIDIAG study) in Mosango, Kwilu Province in the DRC. In that study, consecutive patients with recent-onset neurological disorders admitted to Mosango General Referral Hospital had been recruited between 2012 and 2015 [28]. None of the OAE cases received neither ivermectin nor anti-epileptic drugs. Blood and CSF samples were stored at −80 °C until analysis. 

Stored CSF from the following European controls was also tested: seven persons with epilepsy, four persons with another neurological disorder such as dementia, multiple sclerosis, meningitis, Parkinson’s disease. 

### 4.1. MSD Neuro-Inflammatory Panel

The following cytokines were coated on the wells of a meso-scale discovery (MSD) custom plate: IL-6, TNF-α, IL1-β, IL-5, IL-4, IL-13, CCL3 (Mip-1α), VEGF-C, VCAM-1. 

### 4.2. Statistical Analysis

Data were processed and analysed using IBM SPSS statistics. Differences between the different groups were calculated using Kruskal–Wallis test and if this was significant, pairwise comparison was performed using a Bonferroni correction. *p*-values below 0.05 were considered significant.

### 4.3. Litterature Review 

To review the literature, we searched for articles indexed in PubMed and Google scholar up to 31 January 2021. Search terms included “cytokines”, “nodding syndrome”, “epilepsy and onchocerciasis”, “immune response”.

## 5. Conclusions

In conclusion, based on the results of our pilot study and review of the literature, it is not possible to determine the causative role of neuro-inflammation in OAE including nodding syndrome. Observed cytokine profiles may be the consequence of long standing epilepsy, concomitant infections and malnutrition. Ideally cytokine levels should be determined in a prospective study, and serum and CSF should be examined at the time of onset of the first seizures. 

## Figures and Tables

**Figure 1 pathogens-10-00310-f001:**
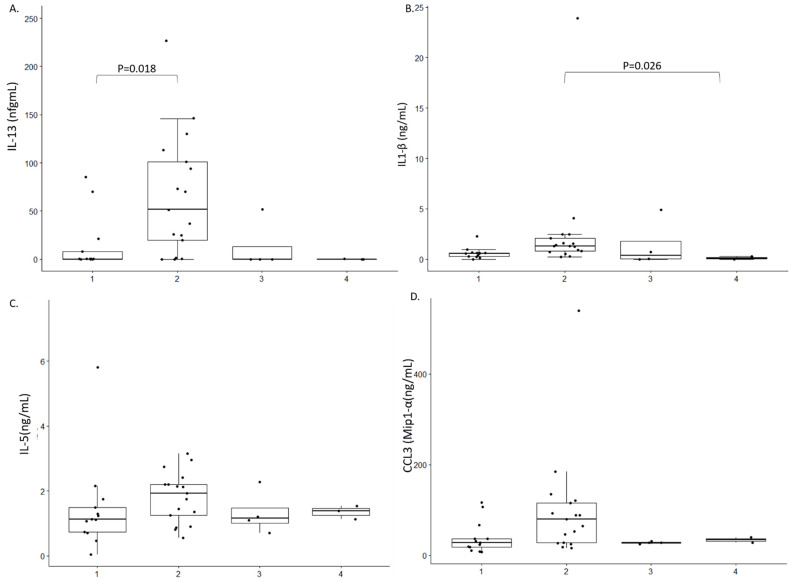
Cytokine expression profile in the study population. 1: Persons with OAE 2: African controls 3: Europeans with a neurological condition but no epilepsy 4: Europeans with epilepsy. Only difference between 2 groups with a *p*-value < 0.05 are shown. (**A**). IL-13 expression in ng/mL; (**B**). Il-1β expression in ng/mL; (**C**) IL-5 expression in ng/mL; (**D**). CCL3 expression in ng/mL.

**Figure 2 pathogens-10-00310-f002:**
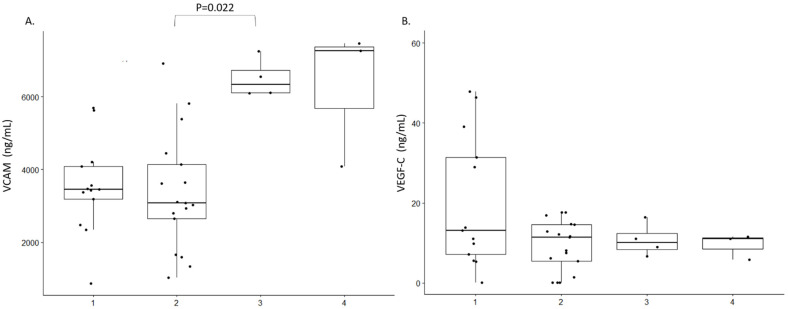
Vascularisation factor expression profile in the study population. 1: Persons with OAE 2: African controls 3: Europeans with a neurological condition but no epilepsy 4: Europeans with epilepsy. Only difference between 2 groups with a *p*-value < 0.05 are shown. (**A**). VCAM expression in ng/mL; (**B**). VEGF-C expressions in ng/mL.

**Table 1 pathogens-10-00310-t001:** Characteristics of 13 persons with OAE (11 from Maridi, South Sudan and two from Mosango, DRC *).

No.	Age (Years)	Sex	ASO (Years)	Seizure Frequency (Monthly)	Generalized TCS	NS	Mf Density (Per SS)	Cognitive Impairment	Onchocerca Skin Lesion	Muscle Wasting
1	17	F	14	1	YES	NO	99.5	NO	NO	NO
2	11	M	5	150	YES	YES	122	YES	YES	YES
3	18	M	6	120	YES	YES	117	YES	YES	YES
4	9	M	7	120	YES	YES	84	YES	YES	YES
5	16	M	4	90	YES	YES	86	YES	YES	YES
6	28	F	1	<1	NO	NO	118.5	NO	NO	NO
7	12	M	10	4	NO	YES	102	NO	NO	NO
8	15	F	7	24	YES	YES	85	NO	YES	NO
9	19	F	15	12	YES	YES	110	NO	NO	NO
10	11	F	5	8	YES	NO	105	NO	NO	NO
11	24	M	14	90	NO	NO	108	NO	NO	NO
12 *	8	M	NK	NK	NK	NK	NK	NK	NK	NK
13 *	28	F	NK	NK	NK	NK	NK	NK	NK	NK

ASO = age of seizure onset, TCS = tonic-clonic seizures, NS = nodding seizures, Mf = microfilarial load, SS = skin snip, NK = not known.

## Data Availability

The datasets generated during the current study are available from the corresponding authors on reasonable request.

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
