# Peer review of "Cytokines and Onchocerciasis-Associated Epilepsy, a Pilot Study and Review of the Literature"

_pathogens, 2021, doi:10.3390/pathogens10030310_

Round 1

Reviewer 1 Report

This initial report on findings of cytokine and chemokine levels in the CSF of patients with OAE seems premature. At this point, only 13 patients have been tested and in some cytokine measurements up to 23% of samples were found to be outliers. This is concerning and suggests greater heterogeneity in cytokine levels and therefore increased sample sizes are needed before any conclusions are made. In that context, this paper feels premature and in need of additional samples to ensure robust analyses. 

My other concerns are that the authors tested 10 cytokines but only provided data to the readers on six of these. The other four should be presented as well. 

Further, line 95 states that the authors tested for MIP1b (CCL4) but the graph presented shows MIP-1a (CCL3). Please ensure all cytokines and chemokines are presented in their standard nomenclature (CCL, CC, or CXC) with common or traditional names given in brackets. For example, CCL3 (MIP-1a).

Finally, the authors cite references showing that CXCl10, CXCL13, and CCL2 are the most important indicators of neuroinflammation in the pediatric population (lines 132-134) yet the authors, who are attempting to discover evidence of neuroinflammatory processes in these patients, do not include these cytokines in their analyses. The authors should evaluate the CSF samples for these markers as well.

Again, this paper feels premature and it would benefit from an expanded cytokine panel and additional samples.

Reviewer 2 Report

The manuscript present interesting preliminary data showing downregulation of IL-13 in the CSF of persons with OAE. This downregulation may contribute to the pathophysiology of seizures in OAE.

However, this manuscript contains major concerns.

Figure 1, panel II-1b, IL-5, and Mip1a should bar graphs that identify which groups are compared.

Please provide the rationale of including persons with Alzheimer's disease in this study. This data should not be included as it detract for the main purpose of the study unless there is group of persons with Alzheimer's disease and epilepsy.

Author Response

Point 1: The manuscript present interesting preliminary data showing downregulation of IL-13 in the CSF of persons with OAE. This downregulation may contribute to the pathophysiology of seizures in OAE.However, this manuscript contains major concerns.Figure 1, panel II-1b, IL-5, and Mip1a should bar graphs that identify which groups are compared.

Response to point 1 : We compared all groups. No significant difference was observe in the cytokine profile of Mip1a and Il-5 but we observed a significant difference between European and African controls in the expression profile of IL-1β (P=0.011). We now report in the results “A higher level of IL1-β was observed in Africans with non-OAE neurological disorders compared to Europeans with epilepsy (p=0.039). No significant difference was observe within the 5 groups for IL-5 and CCL3 (Mip1-α). Moreover, we did not observe any significant difference after multiple comparisons in the vascularisation expression profile (VCAM and VEGF-C) between cases and controls” When no differences were observed between groups we did not show the p value not to overload the figure. We now mention in the footnote of all figures that “P-values are only shown when there was a significant difference between groups (p value < 0.05).”

Point 2:Please provide the rationale of including persons with Alzheimer's disease in this study. This data should not be included as it detract for the main purpose of the study unless there is group of persons with Alzheimer's disease and epilepsy.

Response to point 2: In a post-mortem study of children who died with nodding syndrome by Pollanen et al it was suggested that OAE is a neurodegenerative diseases. In the brains examined in their post mortem study tau deposit as in Alzheimer disease were observed (Pollanen et al., 2018). Therefore, we included an Alzheimer comparison group to compare the cytokine expression profile of person with OAE and Alzheimer disease. Therefore, we propose to keep this group in the paper. We now state in the methods “Three persons with Alzheimer’s disease were included as additional controls because tau deposits, also present in Alzheimer’s disease, had been observed in brains of children who died with nodding syndrome.”

Round 2

Reviewer 2 Report

Figure 1 is still confusing. The authors stated that "P-values are only shown when there was a significant difference between groups" Which groups? Some of the "*" are misplaced.

The Alzheimer data should be removed from this report; conclusions would benefit from better powered studies.

Author Response

Point 1 : Figure 1 is still confusing. The authors stated that "P-values are only shown when there was a significant difference between groups" Which groups? Some of the "*" are misplaced.

Response to point 1: We now adapted the figure. We removed the "*" which were confusing and replace by "dots" representing each sample. The P-value are indicated in the graph withing groups that have a statistical difference.

Point 2: The Alzheimer data should be removed from this report; conclusions would benefit from better powered studies.

Response to point 2 : We have now removed Alzheimer data from the study and adapted the statistical analysis.

Round 3

Reviewer 2 Report

The authors have been responsive to my comments.